# Strategies and responses to the effects of Climate Change on health systems in Sub-Saharan Africa: A scoping review protocol

Chancy Skenard Chimatiro[1,2]*, Solange Mianda[1], Precious Hajison[3,4],
Martina Lembani[1]

**1** University of the Western Cape, School of Public Health, Bellville, South Africa, **2** Department of Administration, Machinga District Health Office, Liwonde, Malawi, **3** University of Pretoria, School of Health Systems and Public Health, Hatfield, Pretoria, South Africa, **4** Department of Research, PreLuHa Consultancy, Zomba, Malawi

\* cchimatiro@gmail.com

## Abstract

### Background

Climate change is recognized as the greatest global health threat of the 21st century. Projections suggest that the Sub-Saharan African region will face more consequences of climate change than any other region globally. The health systems within the region have been affected by the negative effects of climate change. Mapping strategies and responses used in the region to address the effects of climate change on health systems in Sub-Saharan Africa could be a starting point for understanding evidence-based decision-making to inform best practices.

### Methods

This scoping review will follow the methodological framework by Arksey & O'Malley. A wide range of databases will be searched to identify articles published on the strategies and responses to the effects of climate change on the health systems in Sub-Saharan Africa. Only peer-reviewed articles (original quantitative and qualitative studies, mixed methods, systematic reviews, editorials, and commentaries) published in English Language between 2010 and 2024 will be reviewed. All Book chapters and the grey literature (dissertations, conference proceedings, abstracts, reports) and publications primarily focusing on climate change strategies and responses without effects on health systems will be excluded. Covidence software will be used during study selection, data extraction, and summary. Deductive thematic analysis will be performed using predetermined themes from the objectives.

**Data availability statement:** All relevant data from this study will be made available upon study completion.

**Funding:** The author(s) received no specific funding for this work.

**Competing interests:** The authors have declared that no competing interests exist.

## Dissemination

The results of this scoping review will be disseminated at local and international research conferences. Furthermore, the findings will be published in open-access journals targeting different audiences. The findings will also be shared with the Ministry of Health in Malawi for possible policy considerations.

## Introduction

Climate change (CC) is recognized as one of the greatest global health threats of the 21st century. The World Health Organisation (WHO) projections indicate that between the years 2030 and 2050, CC will cause an additional 250,000 deaths per year from malnutrition, malaria, diarrhea, and heat stress [1]. It further indicates that CC environmental stressors, such as air pollution, hurricanes, and floods, will contribute to premature deaths worldwide [1]. The word climate change has been defined and described in various ways by stakeholders. For instance, the United Nations Framework Convention on Climate Change (UNFCCC) defines it as a transmute of climate ascribed directly or indirectly to the activity of humans that modifies the components of the atmosphere as well as natural climate variability observed over a comparable period, usually one or more decades [2]. On the other hand, the Intergovernmental Panel on Climate Change (IPCC) describes it as a state of the climate identified statistically by changes in the mean or the variability of its properties that persist for an extended period, typically decades or longer [3]. However, the most important variables that provide a proper description of the concept of CC are the time involved, the level of variability that the change took, the extremities, and the impact of such variability on humans and the ecosystem [4]. The causes of CC are mainly attributed to a significant increase in anthropogenic greenhouse gas emissions into the atmosphere [5,6].

Evidence indicates that many countries have been affected by the effects of CC globally, which have reduced their capacity to achieve the United Nations Sustainable Development Goals, especially goal number 3, of ensuring good health for all by the year 2030 [7]. However, the effects of CC are disproportionately felt worldwide [8], with some regions experiencing more adverse effects than others. Low and middle-income countries are more vulnerable to the effects of CC and require meaningful actions to reduce their impact [9–11].

The Sub-Saharan African region is facing advanced consequences of CC than any other region because of its low adaptive capacity due to the limited resources [12–14]. It is reported that over 40 million people were affected by floods between 2010 and 2020, while drought contributed to 88% of all CC related disasters in Sub-Saharan Africa [15,16]. The region has weak infrastructure, inadequate human resources, weak institutions, and economic dependency [17,18], making it more vulnerable with insufficient capacity to respond to the effects of CC. The most worrisome situation is that the impact of CC has not spared any sector. And the most affected sectors include agriculture, water, environment, and health [19]. Some of the

more pronounced effects of CC within the region include droughts, increased heatwaves, floods, cyclones, and infectious diseases [18,20,21].

Infectious diseases like malaria, for instance, are significantly influenced by the CC as the change in rainfall patterns and temperatures creates favourable environmental conditions for the spread of such diseases [22]. Another study reported on the geographical shift seasonality of malaria incidence due to CC, which will put additional lives at risk of the endemic [23]. A similar situation has been reported on waterborne diseases such as cholera [24]. It is not surprising that Malawi experienced the worst cholera outbreak associated with CC in the year 2022, which claimed over 1,768 lives [25–27]. Interestingly, the outbreak was aggravated by Cyclone Freddy, associated with CC, thereby increasing pressure on the already resource-constrained health system [25].

Health systems are considered dynamic structures composed of interconnected parts functioning together to improve health outcomes with balanced interactions between their components [28,29]. According to the WHO, a health system is described as consisting of all organizations, people, and actions whose primary intent is to promote, restore, or maintain health [24]. It has six building blocks (governance, service delivery, health financing, medicinal products, human resources, and health information) functioning together to improve health outcomes [30]. The health systems in Sub-Saharan Africa are characterized by a mix of public and private providers, with the majority being public facilities. For instance, Malawi's health system comprises public, private for-profit, and private not-for-profit facilities, with public facilities providing free health services at the point of contact [31]. However, the health systems within the region face similar challenges such as inadequate human resources, drug shortages, fewer training opportunities, limited treatment options, as well as donor dependence due to funding constraints [6,32].

Being a complex structure, health systems in Sub-Saharan Africa are overwhelmed with the effects of CC, as the occurrence of any extreme weather event affects the operation and performance across the building blocks [33]. For example, floods disrupt service delivery as roads are washed away, limiting access to health facilities [34]. Therefore, the WHO is advocating for the health systems' preparedness to protect them from the wide range of impacts of CC by building better, more climate-resilient, and environmentally sustainable health system structures [6].

The effects of CC on health systems have been clarified and categorized in terms of changes in volume and patterns of health service demand from increased morbidity and mortality [32,35,36], reduced capacity to meet the primary and public health needs due to interruption of vital components of the health system such as infrastructure or equipment damage, increased pressure on human resource [37–40] and interruption of other health-related services such as access to clean water, sanitation, food insecurity, disrupted energy supply and communication [41]. These effects limit access to health services, reduce the quality of service delivery, interrupt the supply chain system, create a high demand for human resources, and increase pressure on infrastructure and medical equipment maintenance, which affect the overall operations of the health systems [42]. In addition, heat-related illnesses, increased infectious diseases, malnutrition, water scarcity, and mental illness are CC effects that increase demand for health systems service delivery and resources [43]. The most notably vulnerable groups affected are women, children, older adults, communities of marginalized identity, displaced persons, people with pre-existing health conditions, and those living in poverty who are in the majority within the sub-Saharan region as compared to other regions of the world [44,45].

In response to the effects of CC, health systems use several approaches and strategies to respond to and mitigate its impact. Formulation of national health climate change adaptation plans and health policies [46] is among some of the approaches and strategies being advocated and employed. Furthermore, health systems are encouraged to develop strategies for delivering essential health services, assess the vulnerabilities and capacities of the health systems, and enhance early warning surveillance systems targeting climate-sensitive diseases and their risks [43,44]. Others have suggested that the health system's response to CC can be strengthened through the formulation of interdisciplinary collaboration with relevant sectors and stakeholders [42,45].

Challenges that arise from the effects of CC on health systems require an urgent response to minimize its impact and improve overall performance. The countries within the Sub-Saharan region are party to various international agreements on CC aimed at reducing its effects on human health. One of the legally binding agreements is the 2015 Paris International Treaty on CC, adopted by 196 countries [47]. The agreement aimed at holding the increase in the global average temperature to below 2°C above pre-industrial levels and pursuing efforts to limit the temperature increase to 1.5°C above pre-industrial levels. Furthermore, the first aspiration for the African Agenda 2063 is to create a prosperous Africa based on inclusive growth and sustainable development [48]. Under this aspiration, many countries are aiming to promote Universal Health Coverage (UHC) by ensuring the provision of quality healthcare services for all, enhancing inclusive growth, and reducing vulnerability to CC, which requires a holistic approach.

There is growing attention to tackling and reducing CC effects on health systems in the Sub-Saharan African region. Therefore, mapping strategies and responses used to address the effects of CC on health systems in Sub-Saharan Africa is a starting point for understanding evidence-based decision-making to inform policy formulation and practices within the region. Hence, this scoping review seeks to identify strategies and responses to the effects of CC on the health system within the Sub-Saharan African context. This scoping review is part of a large study conducted to assess health systems preparedness and response to the effects of CC extreme weather events in Malawi.

## The aim and objectives

This scoping review aims to establish and summarise the available evidence about strategies and responses to the effects of climate change on health systems in Sub-Saharan Africa.

### Objectives

1. To identify available strategies and responses used to manage the effects of climate change on health systems within Sub-Saharan Africa.

2. To identify implementation challenges with the strategies and responses in addressing the effects of climate change on health systems within Sub-Saharan Africa.

## Methods

We plan to conduct this study between January to March 2025, and results will be made available by the end of March 2025. The scoping review will apply the methodological framework by Arksey & O'Malley updated by Peters [49,50] based on the following five stages: (i) research question identification, (ii) review of relevant studies, (iii) study selection, (iv) data charting and (v) summarizing and reporting results. In addition, this review will adhere to the Preferred Reporting Items for Systematic reviews and Meta-Analyses (PRISMA) guidelines for conducting and reporting scoping reviews [51,52]. This will offer standardized reporting criteria to enhance transparency and replicability of the findings [53].

**Stage i: Research question identification.** In this scoping review, the guiding research question will be, "What are strategies and responses to the effects of climate change on health systems in Sub-Saharan Africa?". The research question was developed considering the Population, Concept, and Context (PCC) framework [54] as shown in Table 1 below.

**Stage ii: Identifying relevant studies.** We will consult a faculty of Health Sciences Librarian at the University of the Western Cape to develop a working framework for search strategies. The criteria to be followed will be categorized according to the PCC. Then, a comprehensive literature search of studies done and reported on the strategies and responses to the effects of CC on health systems in Sub-Saharan Africa will be conducted in several databases, including PubMed, CINAHL, Google Scholar, and Medline. Full-text articles will be obtained using search strings containing

**Table 1. Population, concept, and context.**

| Population | Concept | Context |
|---|---|---|
| Health systems, service delivery, primary health care, human resources for health, medicinal and vaccine products, leadership and governance, Information for health, health financing, health services | Strategies and responses, climate change | Sub-Saharan Africa, Central Africa, East Africa, Southern Africa, and West Africa |

keywords using the 'AND' and 'OR', Boolean operators. Some of the keywords to be used during the search include "Climate Change"

- "extreme weather events"- "climate variability" AND "health system"- "healthcare system"-"health service"- "public health" AND – "strategies"- "responses"- "adaptation"- "mitigation"- "resilience"- "preparedness" AND - "Sub-Saharan Africa" – "eastern and central Africa" – "southern Africa" – "western Africa. This will be developed in consultation with the faculty Librarian. The final list of articles will be exported to the EndNote referencing manager. This stage will be conducted between January and March 2025.

**Stage iii: Selection of eligible studies.** We will export the list of articles from the EndNote referencing manager to Covidence, a software for conducting scoping and systematic reviews. Only peer-reviewed articles (original quantitative and qualitative studies, mixed methods, systematic reviews, editorials, and commentaries) published in the English language between 2010 and 2024 will be reviewed. We will only consider articles done in the English Language as the study does not have enough funds to hire language translators for other official communication languages in the Sub-Saharan African region. Furthermore, this scoping review will target articles published between 2010 and 2024, as the period covers a decade, which is in line with the CC definition [2,3]. All book chapters and the grey literature (dissertations, conference proceedings, abstracts, reports) and publications primarily focusing on CC strategies and responses without effects on health systems will be excluded.

Two stages will be implemented during the study selection process. The first stage will involve the determination of study eligibility based on the stated inclusion criteria by reviewing study titles and abstracts by two reviewers. The second stage of the selection process will involve reviewing the identified articles in full. All these stages will be done using the Covidence software blindly, and the results will be compared at the end. Any differences that may emerge during any of the two stages, the reviewers will discuss to reach a consensus. If unresolved, the third reviewer, who will act as a tie-breaker, will be invited to review and resolve the differences.

**Stage iv: Data charting.** We will create an Excel data extraction form where all targeted information will be entered. A descriptive-analytical method will be used to extract information from the identified studies. Data charting and extraction will be done with the help of the Covidence software. The relevant information, such as the author(s) name(s), type of study, the study setting, the strategies and responses used, and implementation challenges, will be extracted and entered on the data Excel extraction form for analysis. The process will be completed by the end of March 2025.

**Stage V: Summarizing and reporting the results.** This scoping review will follow the work of Levac et al (2010) and Nelson et al (2015) in presenting a numerical overview of the amount, type, and distribution of the included studies by country [55,56]. Deductive thematic analysis will be performed using predetermined themes from the objectives. We expect the results to be ready by the end of March 2025.

## Review registration

This scoping review protocol has been registered with Open Science Framework at https://doi.org/10.17605/OSF.IO/FQWAP.

**Ethics**

The scoping review does not involve the use of human subjects; as such, no ethics clearance is required. However, being a part of a larger study, the ethics clearance was approved by the Biomedical Research Ethics Committee (BMREC) of the University of the Western Cape.

## Discussion

This scoping review will serve as a base for collecting evidence about strategies and responses to the effects of CC on health systems in the Sub-Saharan African region. The information obtained will assist in understanding and assessing health systems preparedness and responses to the effects of CC extreme weather events in Malawi, which is one of the countries in the Sub-Saharan Africa region. The health systems within the region are characterized by and facing similar CC challenges such as floods, drought, disease outbreaks, and increased heat [35]. Although these CC consequences are felt in both rural and urban areas, however, there are distinct differences between them. Climate-sensitive diseases like waterborne diseases, malnutrition, and displacement are more pronounced in rural areas [43], while heat stress, air pollution, and infrastructure strain are more reported in urban areas [32], especially in non-informal settlements.

Ideally, the generated evidence will help strengthen and improve the health system's performance in Malawi to effectively respond to the effects of CC extreme weather events through the continuous provision of quality health services. Furthermore, we expect the results to inform policy consideration in the future. We anticipate that the results of this scoping review will agree with a study conducted on the global health systems response to climate change adaptation, where health policy and planning, setting up of early warning systems, surveillance, and monitoring were among the reported responses [53].

Our scoping review has some limitations. One of the known limitations is the language barrier, as some countries in Sub-Saharan Africa do not use English as their official language, and the study cannot hire a language translator due to inadequate funds. However, we will ensure that a wide search, using various search engines, is conducted and that all available articles are extracted. In addition, the exclusion of book chapters and the grey literature (dissertations, conference proceedings, abstracts, reports), and publications primarily focusing on CC strategies and responses without effects on health systems will limit the comprehensiveness and depth of evidence to be found. We plan to minimize this through extensive consultations with key stakeholders in policy planning at the MoH headquarters.

The results of this scoping review will be disseminated at local and international research conferences. Furthermore, the findings will be published in open-access journals targeting different audiences. The findings will also be shared with the Ministry of Health in Malawi for possible policy considerations and the University of the Western Cape Library for future reference.

## Conclusion

The need to address challenges posed by CC on health systems in the Sub-Saharan Africa region cannot be overemphasized. Prompt efforts should be made to ensure that the effects of CC on health systems are minimized for the region to achieve the Sustainable Development Goals [7] and the desired African Agenda 2063 [48]. More research should be conducted to understand and determine the best strategies and responses to be used to reduce the effects of CC on health systems within this region.

## Supporting information

**S1 File. Table 1. Full search strategies for the electronic databases.**
(DOCX)

**S1 Checklist.  PRISMA-P (Preferred Reporting Items for Systematic review and Meta-Analysis Protocols) 2015 checklist: recommended items to address in a systematic review protocol\*.**
(DOCX)

## Author contributions

**Conceptualization:** Chancy Skenard Chimatiro, Martina Lembani.

**Methodology:** Chancy Skenard Chimatiro, Solange Mianda, Martina Lembani.

**Project administration:** Chancy Skenard Chimatiro.

**Resources:** Chancy Skenard Chimatiro.

**Supervision:** Solange Mianda, Martina Lembani.

**Validation:** Chancy Skenard Chimatiro, Solange Mianda, Precious Hajison, Martina Lembani.

**Visualization:** Chancy Skenard Chimatiro, Solange Mianda, Precious Hajison, Martina Lembani.

**Writing – original draft:** Chancy Skenard Chimatiro.

**Writing – review & editing:** Chancy Skenard Chimatiro, Solange Mianda, Precious Hajison, Martina Lembani.

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
