## [Decision Letter · Decision Letter 0]

18 Jun 2025

Dear Dr. Chimatiro, 

Thank you for submitting your manuscript to PLOS ONE. After careful consideration, we feel that it has merit but does not fully meet PLOS ONE’s publication criteria as it currently stands. Therefore, we invite you to submit a revised version of the manuscript that addresses the points raised during the review process.

Overall, this is an important and interesting study! Please see the revision notes below.

We look forward to receiving your revised manuscript.

Kind regards,

Jobst Augustin, Associate Professor/Senior lecturer

Academic Editor

PLOS ONE

Journal Requirements:

- https://doi.org/10.3390/ijerph15030538

-https://kmco.co.ke/wp-content/uploads/2023/01/VOLUME-2-ENVIRONMENTAL-BOOK-FINAL.pdf

In your revision ensure you cite all your sources (including your own works), and quote or rephrase any duplicated text outside the methods section. Further consideration is dependent on these concerns being addressed.

3. In the online submission form, you indicated that: 

“The data will be available upon reasonable request from the corresponding author”

**Additional Editor Comments** :

*The planned study is a relevant piece of work. The scoping review protocol is well written and only requires minor revisions:*

- For clarity, the term 'Sub-Saharan Africa' should be defined at the beginning. The fact that the focus is on Malawi is only mentioned later.

- The consequences of climate change on the health system are mentioned, but not the associated costs. Are these not relevant here?

- The implementation schedule should be specified in more detail under Methods.

Reviewers' comments:

Reviewer's Responses to Questions

**Comments to the Author**

1. Does the manuscript provide a valid rationale for the proposed study, with clearly identified and justified research questions?

Reviewer #1: Yes

2. Is the protocol technically sound and planned in a manner that will lead to a meaningful outcome and allow testing the stated hypotheses?

Reviewer #1: Partly

3. Is the methodology feasible and described in sufficient detail to allow the work to be replicable?

Reviewer #1: Yes

4. Have the authors described where all data underlying the findings will be made available when the study is complete?

Reviewer #1: Yes

5. Is the manuscript presented in an intelligible fashion and written in standard English?

Reviewer #1: Yes

You may also provide optional suggestions and comments to authors that they might find helpful in planning their study.

*Reviewer #1: Thank you for sharing this interesting scoping review protocol. The topic you have chosen is both important and highly relevant. As you move forward with writing the full review, here are a few aspects that you might consider:*

- Include specific numbers and relative risks regarding the prevalence/ incidence of relevant negative health outcomes in your research area Sub-Saharan Africa (to highlith the importance).

- "heat stress" as environmental stressor is not the only cause for premature deaths (due to CC). Air pollution, hurricanes, and floods could also play an important role? (You can clarify your sentence: "CC will cause an additional 250,000 deaths per year from malnutrition, malaria, diarrhea, and heat stress")

- In your research environment, are there differences between the consequences of CC in rural and urban areas? (Perhaps this is also a relevant topic for your discussion.)

- Does climate change only affect the infrastructure of health systems? It may also have an impact on human resources (which are particularly lacking in Malawi) and, in the worst case, even lead to cumulative effects of climate change.

- You could describe the role of infectious disease more detalied (Malaria and the Cholera outbreak in 2022).

- Briefly describe the health care system in Sub-Saharan Africa/ Malawi and it’s challenges (less human resource, less training opportunities, limited treatment options ...) as well as the economic situation in a global context (dependency of other countries etc.) in order to understand why research is important for Sub-Saharan Africa/ Malawi. CC seems to be a major problem that adds to the structural and already existing fundamental problems.

- Since scoping reviews do not formally assess the methodological quality of the included studies, their results are not directly applicable for evidence-based recommendations or policy-making. However, they provide a valuable basis for structuring the research field and identifying existing gaps in the evidence.This should be mentioned in the limitations. (Your sentence: "The findings will also be shared with the Ministry of Health in Malawi for possible policy considerations and University of the Western Cape Library for future reference" should be worded more carefully.)

- Check consistency in terminology (e.g., climate change vs. CC).

**Do you want your identity to be public for this peer review?** For information about this choice, including consent withdrawal, please see our Privacy Policy

Reviewer #1: No

---

## [Author Response · Author response to Decision Letter 1]

25 Jun 2025

We thank the reviewer for taking time to provide an oversight and highlight some of areas we missed. The views and all suggestions have been included in the edited manuscript. We are looikng forward to the outcome of the revised manuscript and responses provided as soon as possible.

---

## [Editor Report · Decision Letter 1]

22 Jul 2025

Strategies and responses to the effects of Climate Change on health systems in Sub-Saharan Africa: A scoping review protocol

PONE-D-24-57662R1

Dear Dr. Chancy Skenard Chimatiro, 

We’re pleased to inform you that your manuscript has been judged scientifically suitable for publication and will be formally accepted for publication once it meets all outstanding technical requirements.

Kind regards,

Jobst Augustin, Associate Professor/Senior lecturer

Academic Editor

PLOS ONE

---

## [Editor Report · Acceptance letter]

PONE-D-24-57662R1

PLOS ONE

Dear Dr. Chimatiro,

I'm pleased to inform you that your manuscript has been deemed suitable for publication in PLOS ONE. Congratulations! Your manuscript is now being handed over to our production team.

Kind regards,

on behalf of

Dr. Jobst Augustin

Academic Editor

PLOS ONE